# The Short-Term Effects of an Exercise Protocol Incorporating Blood Flow Restriction and Body Cooling in Healthy Young Adults

**DOI:** 10.3390/mps8060135

**Published:** 2025-11-05

**Authors:** Andrew J. Stanwicks, Patrick C. Pang, Hannah C. Allgood, Yuho Kim, Yi-Ning Wu

**Affiliations:** 1Department of Physical Therapy and Kinesiology, University of Massachusetts Lowell, Lowell, MA 01854, USA; andrew_stanwicks@student.uml.edu (A.J.S.); patrick_pang@student.uml.edu (P.C.P.); hannah_allgood@student.uml.edu (H.C.A.); yuho_kim@uml.edu (Y.K.); 2UMass Lowell New England Robotics Validation and Experimentation Center, University of Massachusetts Lowell, Lowell, MA 01854, USA

**Keywords:** blood flow restriction, interval training, oxygen consumption, post-exercise metabolism

## Abstract

Blood flow restriction (BFR) and body cooling (BC) have been investigated separately during exercise, but little is known about their concurrent use. This study examined acute metabolic responses, respiratory physiology, and rate of perceived exertion (RPE) during interval training (IT) performed with combined BFR and BC (VASPER ON) compared with IT without BFR and BC (VASPER OFF). It was hypothesized that VASPER ON would elicit greater physiological demands. A total of 7 female and 19 male participants (20.2 ± 2.4 years) completed a 21-min IT exercise. In VASPER ON, the participants wore cuffs that simultaneously applied BFR and BC. Total oxygen consumption (TVO_2_), total carbon dioxide production (TVCO_2_), total breaths (BRTH), and total ventilation (TVE) were measured during exercise (EX) and for 10 min post-exercise (Post-EX). RPE was recorded during EX. During EX, TVE and ventilatory equivalents for both oxygen and carbon were significantly higher in VASPER ON. Post-EX, all variables remained significantly elevated in VASPER ON except for the ventilatory equivalent for carbon dioxide. Sprint interval RPE was significantly lower in VASPER OFF. These findings suggest that concurrent BFR and BC increase post-exercise metabolic and ventilatory demands without attenuating each other’s effect.

## 1. Introduction

The benefits of exercise are well established. From a public health perspective, promoting safe and accessible exercise strategies is critical, as physical activity plays a central role in reducing the burden of chronic disease, improving mental health, and enhancing quality of life across populations. However, not all individuals are able to engage in exercise at intensities sufficient to elicit meaningful physiological adaptation. This may be due to a range of factors, such as medical conditions that limit exercise tolerance, for example, in individuals with persistent post-concussive symptoms [1] and people with multiple sclerosis, and practical barriers like time constraints. This presents a significant public health challenge, as these individuals may be excluded from the broad benefits of regular physical activity.

In response, there is growing interest in developing novel exercise strategies that make physical activity more tolerable and adaptable to individual needs. Techniques such as blood flow restriction training and interval training and modalities like thermal modulation have emerged as promising approaches to enhancing physiological adaptations while reducing exertional demands or enhancing recovery. Exploring the combined or synergistic effects of such methods may represent opportunities which offer innovative ways to extend the benefits of training to populations who might otherwise be unable to participate fully in traditional exercise programs.

Blood flow restriction (BFR) training aims to improve exercise efficiency by simulating the effects of a high-intensity workload through restricting venous and arterial blood flow to the muscles downstream. BFR creates a hypoxic environment, which elicits central and peripheral physiological changes which lead to enhanced neuromuscular [2,3,4,5,6,7,8] and cardiovascular performance [2]. Greater improvements in maximal aerobic capacity were also observed when using BFR with low to moderate intensities of aerobic activity [9,10].

Additionally, interval training (IT) has been recognized as an effective exercise method, combining intense workout periods with either complete rest or active recovery involving light activity [11,12]. Light work during the rest period is thought to prevent perceptions of presyncope [12] and has been shown to cause a greater post-exercise hypotensive response compared to that with other exercises [13]. IT allows for a lower training volume with similar benefits to skeletal muscle and performance-related enzyme responses, such as mitochondrial oxidative capacity, and resting muscle glycogen when compared to continuous exercise programs [11,12,14]. Both BFR and IT are being proposed to improve exercise efficiency, with less exercise in terms of its intensity and time required. While training under BFR, because of the restricted blood flow, people perceive that a higher difficulty (more effort) is required to accomplish the tasks, which undeniably could be a barrier for applications [15].

Exercise inevitably elevates body temperature, regardless of the mode or efficiency of the activity. For individuals with persistent post-concussive symptoms or multiple sclerosis, this rise in temperature may exacerbate symptoms and further discourage participation in exercise. Managing exercise-induced increases in core temperature has therefore become an important focus of research. Body cooling (BC) strategies applied before (pre-cooling), during (per-cooling), or after (post-cooling) exercise have been shown to alleviate discomfort and in some cases enhance performance. For example, per-cooling improves thermal perception and comfort during exercise in temperate environments without altering oxygen consumption [16] or performance outcomes [17]. Post-exercise cooling has also been shown to accelerate the decline in thermal strain after moderate exercise [18] and to reduce delayed-onset muscle soreness for up to four days [19]. Moreover, recent pre-clinical studies indicate that combining cold exposure with exercise may confer additional metabolic benefits, including reduced liver fat deposition and improved lipid profiles, which could have positive implications for human health [20].

While the individual effects of these interventions are beneficial and some effects of combined BFR and BC or IT and BFR have been investigated [11,12,13,21,22,23,24,25], the acute effects of BFR and BC during IT have yet to be established. In our previous study, we studied the long-term effects of combined BFR and BC during moderate-intensity interval training on a recumbent bike in adults with persistent post-concussive symptoms [25]. In this 12-session program of exercise over six weeks, we found improved cognitive function and other functions with less variability. Where the evidence supports the benefit of a combined approach, the potential mechanism underlying this approach and the acute effects it causes it are not clear. A potential explanation may involve hemodynamic alterations during exercise, as BC causes vasoconstriction, which reduces local blood flow, while BFR reduces venous return and induces local hypoxia and metabolite accumulation. Both cooling-induced vasoconstriction and BFR-induced ischemia modulate the local environment of exercised muscles. When combined during exercise, BC may amplify the ischemic effects of BFR by limiting oxygen delivery. Furthermore, BC is also found to enhance the accumulation of lactate [26]. Therefore, the combination of BC and BFR may increase the physiological demands of exercise, as both elicit sympathetic activation, which potentially leads people to perceive greater exertion when exercising under combined BC and BFR.

Despite our speculation, this response may replicate the stimulus of more demanding exercise while remaining within tolerable limits, as the combined application of BFR and BC introduces complexities that are not yet understood. Further research is needed to develop protocols that maximize the hypertrophic benefits of exercise. Therefore, the purpose of this study was to evaluate acute metabolic response, respiratory physiology, and perceptual exertion of IT exercise with BFR and BC in a healthy population compared to those under IT exercise without BFR and BC. We hypothesized that combining BFR and BC would enhance the physiological demands, even when the mechanical output was maintained at a comparable level.

## 2. Materials and Methods

### 2.1. Participants

Twenty-nine university students were recruited for this study. Three participants were excluded because of incomplete data collections caused by scheduling issues. Completed datasets for 26 participants (7 female and 19 male) were analyzed. The inclusion criteria were (1) being aged 18–35 years, (2) having the ability to sign informed consent, and (3) having the ability to ride a stationary bike for at least 30 min. The exclusion criteria were (1) any known neuro-musculo-skeletal injuries within one year prior to recruitment, (2) any cardiopulmonary or cardiovascular diseases, and (3) potential pregnancy. Table 1 summarizes the demographic data of the participants. This study complied with the tenets of the Declaration of Helsinki and was approved by the Institutional Review Board of the University of Massachusetts Lowell (#18–134). All participants gave written informed consent prior to participating in the study. No adverse events were reported. All participants reported exercising regularly at least twice a week and had no experience with BFR or BC. When it was not possible to blind the participants to the exercise protocol, they were not informed of any potential effects that could arise from either of the two protocols they were to perform. Although an a priori power analysis was not performed, a post hoc power analysis using G*Power 3.1 indicated that the sample size of 26 participants yielded 91% power to detect an effect size of 0.69, based on the observed difference in RPE responses between the two conditions.

### 2.2. Instrument

A Vasper system (Vasper, Mountain View, CA, USA) was used in this study. The Vasper system provides BFR through cuffs filled with chilled water. The Vasper system consists of a BC unit and a modified NuStep T5XR, a recumbent cross trainer (Figure 1). The BC unit chilled the water to 42 °F (5.6 °C) and pumped it to the seat pad and cuffs. Along with the modified copper foot pedals of the recumbent cross trainer, the system aimed to keep the body cool during exercise. To create the BFR, chilled water was pumped to the cuffs that exerted compression on the arms (40 mmHg) and legs (65 mmHg) to primarily restrict venous blood flow from the exercised muscles in our study. The pressure values used in this study were based on the previous literature [27] and participant comfort. Laurentino et al. reported that moderate BFR (~50% of the typical pressure used) can elicit strength gains comparable to those achieved with high-intensity training without BFR, while producing less discomfort. They also noted that wider cuffs require lower pressures to achieve comparable levels of vascular occlusion. For instance, approximately 140 mmHg is required when using an 18 cm cuff to induce vascular occlusion. We used a slightly greater cuff width for the legs; therefore, we set the pressure at 65 mmHg for the legs. Although the cuff width for the arms was smaller than that used by Laurentino et al., arm circumference is considerably smaller than that of the thighs; therefore, we selected 40 mmHg for the arms. These selected values were consistent with those used in our previous study [25]. The Vasper system continuously monitors the water temperature and pressure of the cuffs to maintain consistent compression and cooling throughout the exercise protocol. To investigate the differences between with and without using BFR and BC during IT (VASPER ON versus VASPER OFF), the BC and BFR were turned off during VASPER OFF condition.

A portable metabolic measurement system (Cosmed K5, COSMED USA, Inc., Concord, CA, USA) was used to measure breathing rate, ventilation, oxygen consumption rate, and carbon dioxide output rate via the Cosmed software platform, OMNIA (version 1.6.3). The flowmeter and the analyzers of the system were calibrated before each testing day and after it had been on for 45 min. We used a 3 L syringe to calibrate the flowmeter. The remaining procedures included scrubber calibration that zeroed the CO_2_ analyzer, reference gas calibration using the known reference gas (16% O_2_, 5% CO_2_, balance nitrogen), and delay calibration for the breath-by-breath mode used in this study.

### 2.3. Exercise Protocol

Preprogrammed IT exercise provided by the machine was adopted in this study for the two conditions, IT with BFR and BC (VASPER ON) and IT without BFR or BC (VASPER OFF). The Super Six program is a 21-min exercise starting with a 9-min warm-up leading into six 30-s sprint intervals, each followed by a 90-s active rest interval (Figure 2). Each interval had a maximum and minimum wattage range determined by the resistance level of the machine that the participant needed to stay within. To determine the resistance levels, the rate of perceived exertion (RPE) scale (Borg CR-10) was used. In this study, the VASPER ON condition was designated as the primary protocol, and parameters were first established under this condition. These parameters were subsequently applied to the protocol without VASPER BRF and BC in order to assess how outcomes differed when identical parameters were employed across both conditions. The participant had to reach the target RPE scale level of 6 (higher end of moderate intensity) during the sprint intervals in the VASPER ON condition. If the sprint interval was deemed too easy, the next sprint level’s resistance was increased by one level. The resistance levels of the sprint intervals decided in VASPER ON were saved and used for the VASPER OFF condition for each individual participant. VASPER ON was conducted first to determine the appropriate resistance for the moderate-intensity workout, based on the assumption that it would be the more demanding of the two conditions. This approach allowed for adjustments to the sprint intervals to be made and saved, ensuring that participants were able to complete the identical protocol in both conditions. Using feedback displayed on the tablet screen (Figure 1), participants had to generate preset Watts determined by the resistance level. Because the resistance levels remained the same in two conditions, the mechanical output remained comparable. No resistance level changes were made to the warm-up and rest intervals of the protocol. Each participant started the study with VASPER ON followed by VASPER OFF at least one day apart to wash out potential effects caused by the exercise and within one week. The waiting period for the participants in this study ranged from one to seven days, with a median of five days and a mode of seven days. Each participant was asked to complete the two condition sessions at approximately the same time of day. All participants attended both sessions at the same time or within an hour of the same time, except for two participants who had unexpected scheduling conflicts and completed VASPER OFF later. During the VASPER OFF condition, all of the cooling and compressive components were removed. After finishing the 21-min IT (EX), participants were asked to rest on a bed for 10 min (Post-EX). The room where the exercise occurred was air-conditioned, with the thermostat set at 20 °C. However, because the system was centrally controlled within a building, the actual temperature was approximately 25 °C, about 2~5 °C higher than the temperature recommended by the American College of Sports Medicine (ACSM) [28] and the International Fitness Association (IFA) but still within the thermoneutral range. Participants did not express any discomfort or report feeling warm before exercise began. The average humidity was about 52%.

### 2.4. Data Analysis

The physiological responses in the two conditions, including total breaths (BRTH), total ventilation of air exchanged in liters (TVE), total volume of oxygen consumed in milliliters (TVO_2_), total volume of carbon dioxide output in milliliters (TVCO_2_), ventilatory equivalent for oxygen (VE/O_2_), and ventilatory equivalent for carbon (VE/CO_2_), were calculated using Microsoft Excel for the EX and Post-EX. To ensure comparable mechanical work was undertaken in the two conditions, we also compared the sum of work (in Watts). Data normality was assessed using the Shapiro–Wilk test. All dependent variables, except for VE/O_2_ and VE/CO_2_ during EX and TVO_2_ Post-EX, were normally distributed. Paired t-tests were used to compare normally distributed variables between the two conditions (VASPER ON vs. VASPER OFF), while Wilcoxon signed rank tests were used to compare non-normally distributed variables and RPE scale scores. For nonparametric data, the Hodges–Lehmann test was used to estimate the 95% confidence interval of difference (CI). Cohen’s d(d) or the effect size r for nonparametric variables was calculated as an effect size for each comparison. Statistical analyses were performed in SPSS (Version 31) with the level of significance (α) adjusted via Benjamini–Hochberg’s test using Microsoft Excel. Each adjusted α is reported in Table 2.

## 3. Results

The average mechanical outputs over 21 min for the VASPER ON and VASPER OFF conditions were 89,909.56 ± 12,090.75 and 92,052.32 ± 8678.07, respectively. The correlation between the two conditions was statistically significant (r = 0.65, *p* < 0.001). There was no significant difference in mechanical output between the two conditions (*p* = 0.258, CI [−5963.71 1678.19], d = −0.231). These results suggest that participants produced consistent mechanical outputs across both conditions.

In general, oxygen consumption and carbon dioxide production did not differ significantly during EX but did Post-EX. Table 2 presents the *p* values for all 13 paired comparisons conducted for EX and Post-EX, along with their adjusted α and the determination of statistical significance. Specific results for the EX and Post-EX phases are elaborated in the following sections.

### 3.1. Physiological Responses to EX

As shown in Figure 3, significant differences were found in TVE, VE/O_2_, and VE/CO_2_ but not in the other variables during EX. Participants experienced increased TVE during VASPER ON when compared to that in VASPER OFF during EX (872.97 ± 176.14 vs. 800.40 ± 98.87, *p* = 0.011, CI [17.81 127.33], d = 0.54). There was no significant difference in TVO_2_ or TVCO_2_ between the two conditions during EX (19.47 ± 3.68 vs. 19.14 ± 3.08, *p* = 0.493, CI [−0.65 1.31], d = 0.14 and 17.70 ± 3.46 vs. 17.32 ± 2.96, *p* = 0.492, CI [−0.74 1.50], d = 0.14, respectively) in addition to BRTH (654.08 ± 132.58 vs. 618.97 ± 122.05, *p* = 0.057, CI [−1.14 71.34], d = 0.39). We observed significant increases in VE/O_2_ and VE/CO_2_ during VASPER ON when compared to those in VASPER OFF during EX (30.49 ± 4.32 vs. 28.53 ± 3.08, *p* = 0.005, CI [0.58 3,19], r = 0.56 and 33.48 ± 4.44 vs. 31.55 ± 3.29, *p* = 0.011, CI [0.65 3.42], r = 0.5, respectively).

### 3.2. Physiological Responses Post-EX

As shown in Figure 4, statistically significant differences were observed in all the variables Post-EX except for VE/CO_2_. In VASPER ON, the participants experienced significantly increased BRTH Post-EX than those in VASPER OFF (220.92 ± 61.02 vs. 203.22 ± 50.41, *p* = 0.009, CI [6.19 38.89], d = 0.56). Statistically significant increases in TVE, TVO_2_, TVCO_2_, and VE/O_2_ were observed in VASPER ON compared to those in VASPER OFF (203.82 ± 42.96 vs. 162.00 ± 27.58, *p* < 0.001, CI [26.19 58.55], d = 1.06, 8.61 ± 2.48 vs. 7.13 ± 0.97, *p* = 0.001, CI [0.42 1.83], r = 0.64, 8.32 ± 2.20 vs. 6.69 ± 1.09, *p* < 0.001, CI [0.84 2.42], d = 0.83, and 35.26 ± 5.49 vs. 33.24 ± 4.42, *p* = 0.022, CI [0.32 3.73], d = 0.48, respectively). There was no significant difference in VE/CO_2_ between the two conditions Post-EX (36.34 ± 4.56 vs. 35.55 ± 4.34, *p* = 0.282, CI [−0.69 2.28], d = 0.22).

### 3.3. RPE During EX

Figure 5 shows the average RPE between the two conditions during the sprint intervals. The average RPE during sprint intervals was significantly higher in the VASPER ON condition than that in the VASPER OFF condition (6.1 ± 1.3 vs. 5.2 ± 1.3, *p* < 0.001, CI [0.42 1.42], d = 0.69).

## 4. Discussion

This study investigated the acute effects of a moderate-intensity interval training protocol that combined BFR and BC, followed by a 10-min cooling period, a protocol not previously examined. The findings demonstrated no significant differences in oxygen utilization between conventional moderate-intensity interval training and moderate-intensity interval training with BFR and BC. However, during the post-exercise phase, oxygen utilization was significantly higher following the BFR and BC protocol. In other words, excess post-exercise oxygen consumption (EPOC) was more evident in the VASPER ON condition.

During exercise, total ventilation and ventilatory equivalents for oxygen and carbon dioxide were significantly higher in the VASPER ON condition compared to VASPER OFF. Under a comparable exercise intensity, it is understandable that oxygen consumption and carbon dioxide production were physiologically similar between two conditions. The increased ventilation in our study might suggest that exercise under BFR and BC facilitates ventilatory drive regardless of the intensity of the exercise. This is also supported by the RPE reported by the participants in our study. Despite the amount of work (mechanical output/power) generated during exercise being comparable between the two conditions, the RPE was significantly lower in the VASPER OFF condition compared to the RPE in the VASPER ON condition. Previous studies have also found that aerobic exercise with BFR had a significantly higher RPE compared to that in aerobic exercise without BFR [15,23,29], which further support our findings on the impact of BFR on perceived exertion. While applying BC during exercise has been found to improve exercise performance and thermal perception and reduce the RPE [30,31], with the combination of BFR, the cooling effects on the RPE found in the previous literature were not observed in our study. This might be due to the fact that when per-cooling is applied with BFR in the same local area, BC may act synergistically with BFR through cold-induced vasoconstriction. The reduced blood flow caused by BC appeared to amplify the effects of BFR during exercise, as evidenced by the increased ventilation and RPE.

Ventilation increasing without a corresponding increase in carbon dioxide production may be because of the effect of BFR on venous blood flow. BFR cuffs applying a pressure similar to the pressure used in our study have been shown to reduce arterial blood flow to 56% of the baseline [24]. Reduced blood flow changes the composition of venous blood, and BFR increases the amount of CO_2_ in the venous blood [32]. Furthermore, BFR compresses the vasculature proximally to the working muscles, inducing local hypoxia and increasing metabolic stress [33]. The accumulation of CO_2_ caused by BFR leads to a subsequent decrease in blood pH. The resulting hypercapnia and acidosis stimulate peripheral chemoreceptors, which send signals to the respiratory centers in the medulla. This activation increases ventilation to facilitate the restoration of homeostasis. Mendonca et al. hypothesized that increased venous CO_2_ that caused an increase in ventilation in their study also caused an increase in oxygen consumption [24]. However, in our study, the increase in ventilation happened independently of oxygen consumption and carbon dioxide production. While exercising in a cold environment may impose additional stress, local per-cooling techniques have been shown to reduce muscle perfusion and oxygen diffusive capacity without altering total oxygen consumption [34]. Therefore, in our study, BC alone did not seem to increase oxygen demand further.

Despite the potential combined stress caused by BFR and BC, we did not observe any significant increase in TVO_2_ or TVCO_2_ in VASPER ON compared to those in VASPER OFF during exercise. The previous literature also shows diverse findings regarding BFR. Corvino et al. studied many different modes of exercise with and without BFR and found that there was no difference in VO_2_ between continuous BFR exercise, interval BFR exercise, and continuous conventional exercise [22]. Conceição et al. found that there was a significant increase in oxygen consumption and aerobic metabolism over 30 min of continuous moderate-intensity exercise with BFR [21]. Mendonca et al. found BFR created a 10.4% increase in VO_2_ compared to that without BFR during walking [24]. Variations in exercise protocols across studies appear to contribute to inconsistent respiratory and ventilatory outcomes. Even small differences in interval length, exercise intensity, cuff pressure, or the application of BFR cuffs may lead to distinct physiological responses. The potential reason that we did not observe significant differences in the physiological demands during exercise between the two conditions might be due to the relatively lower exercise intensity used in this study, as shown in the oxygen consumption during both conditions (19.47 mL·kg^−1^·min^−1^ during VASPER ON and 19.14 mL·kg^−1^·min^−1^ during VASPER OFF). Even though short work intervals with a moderate RPE were used, the resulting intensity in our study seems to be low compared to that in the previous literature. Bond et al. employed a protocol consisting of 1 min of cycling at 90% peak power followed by 75 s of active recovery and reported an average oxygen consumption of ~26 mL·kg^−1^·min^−1^ with a mean RPE of 7 on the modified Borg scale [35]. Harris et al. investigated a training program of 12 sets of 30 s of work followed by 30 s of rest, reporting an average oxygen consumption of 35.0 mL·kg^−1^·min^−1^ and an RPE of 6.8 per set [36]. Other studies found higher oxygen consumption patterns while using maximum-intensity sprints ranging from around 34 mL·kg^−1^·min^−1^ [37] to 47.3 mL·kg^−1^·min^−1^ [38]. All of the aforementioned studies had longer sprint intervals or more sets of sprint intervals compared to our study, which used six sets of 30-s intervals with 90 s of active rest between intervals. RPE is suitable for prescribing intensity in traditional exercise. However, using the RPE to prescribe exercise in BFR exercise seems to create lower oxygen consumption values than desired.

Although no significant differences in the metabolic variables were observed between the two conditions during exercise, the post-exercise metabolic and respiratory variables were significantly higher in the VASPER ON condition compared to those in the VASPER OFF condition, except for the ventilatory equivalent for carbon dioxide. The previous literature has shown that excess post-exercise oxygen consumption (EPOC) elevates with a greater exercise intensity and duration [39,40]. The resistance profile, duration, and mechanical output of the exercise in this study were kept the same for the two conditions. Therefore, the combination of BFR and BC seems to be the primary cause that affected the metabolic and respiratory output we observed post-exercise. The higher breathing rate post-exercise might reflect an adaptive response to elevated EPOC caused by the combination of BFR and BC [41]. Reis et al. also demonstrated that BFR enhances muscle deoxygenation and reduces tissue oxygenation during exercise compared to equivalent workloads without BFR [42]. Reduced oxygen availability during exercise tends to elevate EPOC afterward. These previous findings support our observation that despite identical external workloads, the VASPER ON condition may generate greater oxygen debt, resulting in elevated EPOC compared to that under VASPER OFF. Moreover, our post-exercise results are consistent with previous studies reporting significantly greater metabolic responses to aerobic exercise with BFR compared to those under exercise without BFR [43,44]. The enhanced metabolic rate following BFR may be explained, at least in part, by reduced oxidative stress, as BFR resistance exercise has been shown to ameliorate mitochondrial reactive oxygen species [45]. Furthermore, the previous literature has shown that per-cooling techniques induce glycolytic metabolism [46], which can contribute to EPOC. In our study, the effects of the combination of BFR and BC on metabolic and ventilatory demand became more pronounced post-exercise. The elevated EPOC observed could likely reflect greater lactate accumulation during exercise. These findings further suggest that incorporating BFR and BC imposes greater metabolic demands.

While the findings provide valuable insights, several methodological limitations should be acknowledged. The fixed testing order and lack of counterbalancing may have introduced order effects. Exercise intensity was determined using the RPE, which, while practical, may have introduced variability across participants due to subjective perception. The training status of the participants was not recorded; therefore, we were unable to determine whether training status influenced the observed results. Skin and core temperature, heart rate, and blood lactate measurements were not collected, which limited interpretation of the metabolic and thermoregulatory responses. The lack of a time course assessment of physiological responses during exercise may preclude capturing meaningful physiological patterns that are obscured when using total sums, which should be addressed in future studies with large sample sizes. The modest sample size and unblinded design may also limit the generalizability of the findings.

## 5. Conclusions

The 21-min exercise protocol yielded no significant differences in physiological responses during exercise, except total ventilation, which was higher when the BFR and BC were incorporated into the protocol. However, higher physiological responses were observed during the 10 min after exercising with combined BFR and BC. Our findings suggest that the combination of BC and BFR could potentially create a more intense and metabolically demanding post-exercise state than that without.

## Figures and Tables

**Figure 1 mps-08-00135-f001:**
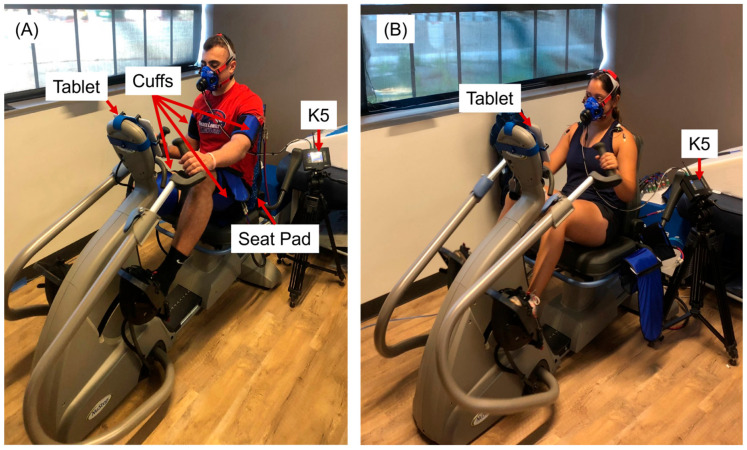
A participant exercising with VASPER ON (**A**) and VASPER OFF (**B**). In the VASPER ON condition, the cuffs were applied to the participant’s upper brachial and femoral regions. The cuff width for the arms and legs is 13 cm and 21 cm, respectively. The system’s tablet displayed the exercise protocol and prompted participants to either sprint or rest. It displayed the target power output that participants were required to sustain by modifying their cadence.

**Figure 2 mps-08-00135-f002:**
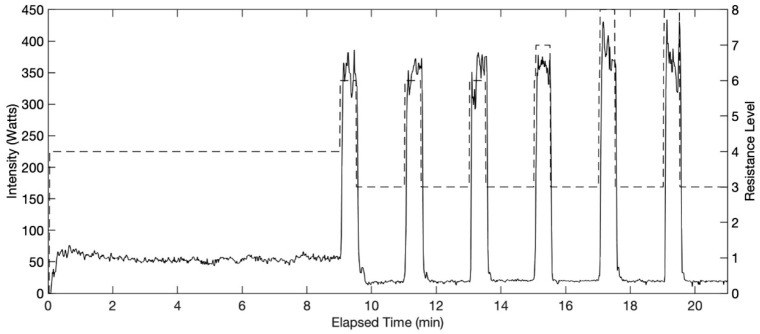
Twenty-one-minute IT included a nine-minute warmup at 30–75 W, six 30-s sprint intervals, and six 90-s rest intervals at 10–30 W between each sprint and after the last sprint. The dashed line indicates the resistance levels set for the intervals in this example IT profile. The resistance levels provided by the exercise machine are on an ordinal scale where higher numbers indicate greater resistance.

**Figure 3 mps-08-00135-f003:**
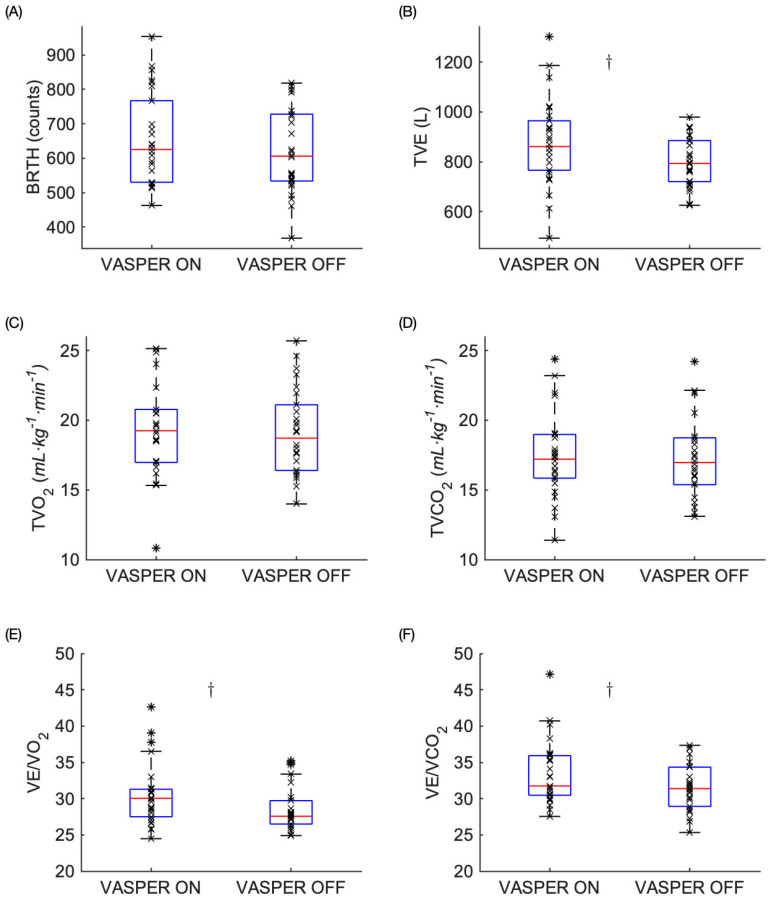
Metabolic and respiratory measures during the EX including BRTH (**A**), TVE (**B**), TVO_2_ (**C**), TVCO_2_ (**D**), VE/O_2_ (**E**), and VE/CO_2_ (**F**). The box represents the interquartile range with the median bar within the box. The whiskers extend to the most extreme data points. The outliers are indicated by the * symbols. The dagger (†) indicates statistically significant differences between two conditions after comparing the *p* values to the adjusted α level using the Benjamini–Hochberg test.

**Figure 4 mps-08-00135-f004:**
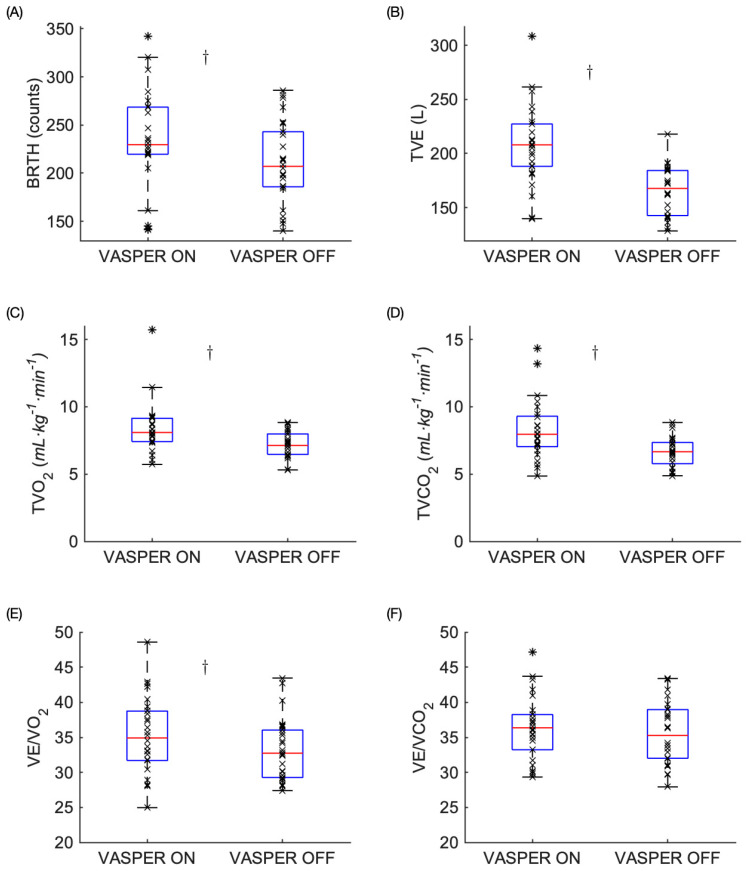
Metabolic and respiratory measures Post-EX including BRTH (**A**), TVE (**B**), TVO_2_ (**C**), TVCO_2_ (**D**), VE/O_2_ (**E**), and VE/CO_2_ (**F**). The box represents the interquartile range with the median bar within the box. The whiskers extend to the most extreme data points. The outliers are indicated by the * symbols. The dagger (†) indicates statistically significant differences between two conditions after comparing the *p* values to the adjusted α level using the Benjamini–Hochberg test.

**Figure 5 mps-08-00135-f005:**
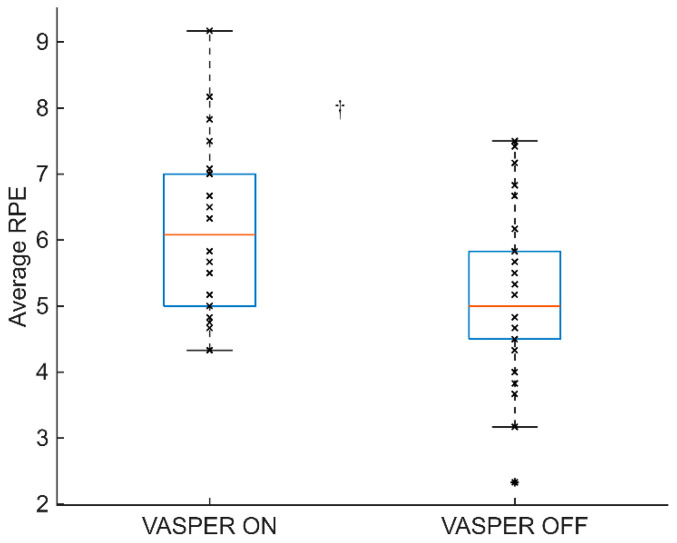
Average RPE of sprint intervals. The box represents the interquartile range with the median bar within the box. The whiskers extend to the most extreme data points. The outliers are indicated by the * symbols. The dagger (†) indicates statistically significant differences between two conditions after comparing the *p* values to the adjusted α level using the Benjamini–Hochberg test.

**Table 1 mps-08-00135-t001:** Demographic data of the participants.

	Male (n = 19)	Female (n = 7)	Total (n = 26)
Age (years)	20.4 ± 2.8	19.7 ± 0.7	20.2 ± 2.4
Height (cm)	175.8 ± 9.3	162.2 ± 5.2	173.9 ± 0.1
Weight (kg)	77.2 ± 8.8	56.3 ± 4.9	71.5 ± 11.8
BMI (kg/m^2^)	25.0 ± 2.6	21.5 ± 3.1	24.1 ± 3.1

**Table 2 mps-08-00135-t002:** Statistical significances of all comparisons compared to the adjusted α.

	*p* Value	Rank	Adjusted α	BH Significance
EX				
BRTH	0.057	10	0.038	no
TVE	0.011	7.5	0.029	yes
TVO_2_	0.493	13	0.050	no
TVCO_2_	0.492	12	0.046	no
VE/O_2_	0.005	5	0.019	yes
VE/CO_2_	0.011	7.5	0.029	yes
RPE	0.001	2.5	0.010	yes
Post-EX				
BRTH	0.009	6	0.023	yes
TVE	0.001	2.5	0.010	yes
TVO_2_	0.001	2.5	0.010	yes
TVCO_2_	0.001	2.5	0.010	yes
VE/O_2_	0.022	9	0.035	yes
VE/CO_2_	0.282	11	0.042	no

BH: Benjamini–Hochberg test.

## Data Availability

The data presented in this study are available upon request from the corresponding author.

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
