# Peer review of "The Short-Term Effects of an Exercise Protocol Incorporating Blood Flow Restriction and Body Cooling in Healthy Young Adults"

_mps, 2025, doi:10.3390/mps8060135_

Round 1
Reviewer 1 Report
Comments and Suggestions for Authors
The main question addressed by the research was the acute effects of an interval training protocol that combined blood flow restriction (BFR) and body cooling (BC), followed by a 10-minute cooling period. The topic is original and relevant to the field by evaluation acute metabolic response, respiratory physiology, and perceptual exertion of interval training (IT) exercise with blood flow restriction (BFR) and body cooling (BC) compared to IT exercise without BFR and BC. The specific gap in the field lies precisely in the fact the of absence of protocols of hypertrophic of BFR simultaneously leveraging the recovery advantages of cooling. Although BFR and BC are well-established strategies with demonstrated benefits for exercise performance, recovery, and rehabilitation, but it is not clear their combined application. address a specific gap in the field.
The manuscript highlights about the acute metabolic responses, respiratory physiology, and rate of perceived exertion (RPE) during interval training (IT) performed with combined blood flow restriction (BFR) and body cooling (BC) compared with IT without BFR and BC; a protocol not previously examined. The topic is presented as original and relevant to the field of Sports Science. This study informs readers about the most appropriate type of material. The methodology of the present study considers instruments appropriate to the experiment carried out. The manuscript's conclusions are duly solid, reflected with scientific evidence and consistent arguments, with objectivity to the research problem presented, with emphasis on the combination of BC and BFR could potentially create a more intense and metabolically demanding post-exercise than without. The study's references are appropriate, up-to-date and consistent with the proposed theme. The component referring to the tables and figures are adequate to the clarifications necessary for the study.
The specific improvements should the authors consider regarding the methodology:
The suggestion that the combination of body cooling (BC) and blood flow restriction (BFR) could potentially create a more intense and metabolically demanding post-exercise than without.
Author Response
- The specific improvements should the authors consider regarding the methodology:
The suggestion that the combination of body cooling (BC) and blood flow restriction (BFR) could potentially create a more intense and metabolically demanding post-exercise than without.
Response: We appreciate the reviewer’s positive comments and helpful suggestions. In response, we added a paragraph in the introduction describing the potential combined effects of BFR and BC on the human body. (Please see Lines 90-98)

Reviewer 2 Report
Comments and Suggestions for Authors
Dear authors, congratulations for your effort and on the selection of such an interesting topic.
Below are several comments for your manuscript:
1.The Abstract states “both oxygen and carbon dioxide relative to ventilation were significantly higher in VASPER ON.” In Results you report VE/VO₂ and VE/VCO₂ were higher in ON (i.e., more ventilation per unit gas). Consider rephrasing for clarity: “ventilatory equivalents for O₂ and CO₂ were higher…”
2.§2.2: “modified cooper foot pedals” Verify that the word cooper is accurate.
3. §2.3: Clarify whether environmental conditions (ambient T, humidity) were controlled, as BC responses are temperature-sensitive.
4. §2.3: Specify washout (≥24 h is stated) and time-of-day control; circadian effects can influence ventilatory/metabolic responses.
5.§2.4: State software used
6. Replace colloquial/vague terms: e.g., “trend” (Results §3.1) → state non-significance everywhere in the text when you are refferring to p>0.05
7. Add a Limitations paragraph summarizing: fixed order, RPE-based titration confound, lack of counterbalancing, absence of VO₂/VE rates in primary reporting, no lactate/core temperature measurements, modest sample size, unblinded protocol.
8. Add power justification and participant training status.
Also, all participants completed VASPER ON first, then OFF, ≥1 day apart. This introduces order/familiarization and expectation effects that may influence ventilatory behavior and Post-EX responses. The absence of counterbalancing is a key limitation. Please (i) acknowledge explicitly; (ii) consider a sensitivity analysis (e.g., first three vs. last three sprints) to evaluate potential familiarization within sessions; and (iii) address in the limitations.
Participant characterization: Provide training status (VO₂peak or habitual activity), and consider reporting BMI; these can modulate responses to BFR and cooling.
9. Height (Total) is reported as 173.9 ± 0.1 cm, (Male: 176.9 ± 9.0, n=19; Female: 162.0 ± 5.3, n=7), verify that total mean and sd 0.1 is correct.
10. For all outcomes, report paired effect sizes with 95% CIs alongside p-values.
11. Discussion section:
Discussion states “no significant differences in oxygen utilization during exercise,” yet later you interpret higher ventilation as reflecting BFR-induced CO₂ handling and ventilatory drive. That is reasonable, but because TVO₂ and TVCO₂ are totals, not rates, the mechanistic argument would be stronger if you showed rate traces or mean rates and end-exercise blood lactate (or surrogate) to support “greater oxygen debt” claims. Consider adding EPOC terminology for Post-EX and, if available, lactate or HR recovery data.
and
You conclude BC did not offset BFR’s effect on RPE. Given the RPE prescription confound, please soften this claim: at minimum, state that under a load titrated in the BFR+BC condition, average RPE during sprints was lower in OFF, and no independent effect of BC can be isolated without a factorial design.
Author Response
Comment 1.The Abstract states “both oxygen and carbon dioxide relative to ventilation were significantly higher in VASPER ON.” In Results you report VE/VO₂ and VE/VCO₂ were higher in ON (i.e., more ventilation per unit gas). Consider rephrasing for clarity: “ventilatory equivalents for O₂ and CO₂ were higher…”
Response 1: Thank you for the suggestion. We have revised the wording as suggested to enhance clarity.
Comment 2: §2.2: “modified cooper foot pedals” Verify that the word cooper is accurate.
Response 2: We corrected the typo.
Comment 3: §2.3: Clarify whether environmental conditions (ambient T, humidity) were controlled, as BC responses are temperature-sensitive.
Response 3: We did not precisely control the environmental conditions; however, both exercise protocols were conducted in the same air-conditioned room and at approximately the same time of day. A corresponding statement has been added to the Methods section as follows.
“The room where the exercise occurred was air-conditioned, with the thermostat set at 20°C. However, because the system was centrally controlled within a building, the actual temperature was approximately 25°C, about 2~5°C higher than the temperature recommended by the American College of Sports Medicine (ACSM) [27] and the International Fitness Association (IFA), but still within the thermoneutral range. Participants did not express any discomfort or report feeling warm before exercise began. The average humidity was about 52%.” (Lines 199-206)
Comment 4: §2.3: Specify washout (≥24 h is stated) and time-of-day control; circadian effects can influence ventilatory/metabolic responses.
Response 4: We have added more information as suggested.
“Each participant started the study with VASPER ON followed by VASPER OFF at least one day apart to washout potential effects caused by the exercise and within one week. The waiting period for the participants in this study ranged from one to seven days, with a median of five days and a mode of seven days. Each participant was asked to complete the two condition sessions at approximately the same time of day. All participants attended both sessions at the same time or within an hour of the same time, except for two participants who had unexpected scheduling conflicts and completed the VASPER OFF later.”
Comment 5: §2.4: State software used
Response 5: We have added the COSMED software platform, OMNIA in 2.2 and Microsoft Excel in 2.4.
Comment 6: Replace colloquial/vague terms: e.g., “trend” (Results §3.1) → state non-significance everywhere in the text when you are refferring to p>0.05
Response 6: The Results section has been revised to remove ambiguous language and enhance precision.
Comment 7: Add a Limitations paragraph summarizing: fixed order, RPE-based titration confound, lack of counterbalancing, absence of VO₂/VE rates in primary reporting, no lactate/core temperature measurements, modest sample size, unblinded protocol.
Response 7: We have addressed limitations of the study in the revised manuscript as follows.
“While the findings provide valuable insights, several methodological limitations should be acknowledged. The fixed testing order and lack of counterbalancing may have introduced order effects. Exercise intensity was determined using RPE, which, while practical, may have introduced variability across participants due to subjective perception. The training status of participants was not recorded; therefore, we are unable to determine whether training status influenced the observed results. Skin and core temperature, heart rate and blood lactate measurements were not collected, which limits interpretation of metabolic and thermoregulatory responses. The lack of time course assessment of physiological responses during exercise may preclude capturing meaningful physiological patterns that are obscured when using total sums, which should be addressed in future studies with large sample sizes. The modest sample size and unblinded design may also limit the generalizability of the findings.” (Lines 391-402)
Comment 8: Add power justification and participant training status. Also, all participants completed VASPER ON first, then OFF, ≥1 day apart. This introduces order/familiarization and expectation effects that may influence ventilatory behavior and Post-EX responses. The absence of counterbalancing is a key limitation. Please (i) acknowledge explicitly; (ii) consider a sensitivity analysis (e.g., first three vs. last three sprints) to evaluate potential familiarization within sessions; and (iii) address in the limitations. Participant characterization: Provide training status (VO₂peak or habitual activity), and consider reporting BMI; these can modulate responses to BFR and cooling.
Response 8: We have added a power justification in the Methods section, provided additional information as suggested, and acknowledged the relevant limitations. We have also revised the statistical analysis and updated the corresponding descriptions in the Methods and Results sections.
“Although a priori power analysis was not performed, a post hoc power analysis using G*Power 3.1 indicated that the sample size of 26 participants yielded 91% power to detect an effect size of 0.69, based on the observed difference in RPE responses between the two conditions.” (Lines 123-126)
We did not assess training status using VO₂peak measurements but instead relied on participants’ self-reported habitual physical activity. This represents a limitation of the study, which we have acknowledged in the limitations section. Additionally, we have included BMI in Table 1 and provided a description of participants’ characteristics alongside the inclusion criteria. As our goal was to evaluate the general applicability of our protocols, training status was not specifically considered. Consequently, we are unable to determine whether training status influenced the observed results, which warrants further investigation.
“All participants reported exercising regularly at least twice a week and had no experience with BFR or BC. When it was not possible to blind the participants to the exercise protocol, they were not informed of any potential effects that could arise from either of the two protocols they were to perform.” (Lines 119-123)
Comment 9: Height (Total) is reported as 173.9 ± 0.1 cm, (Male: 176.9 ± 9.0, n=19; Female: 162.0 ± 5.3, n=7), verify that total mean and sd 0.1 is correct.
Response 9: We appreciated the comment and have made the necessary corrections.
Comment 10: For all outcomes, report paired effect sizes with 95% CIs alongside p-values.
Response 10: Effect sizes, 95% confidence intervals, and p-values have been reported, and the adjusted α values are presented in Table 2.
Comment 11: Discussion section:
Discussion states “no significant differences in oxygen utilization during exercise,” yet later you interpret higher ventilation as reflecting BFR-induced CO₂ handling and ventilatory drive. That is reasonable, but because TVO₂ and TVCO₂ are totals, not rates, the mechanistic argument would be stronger if you showed rate traces or mean rates and end-exercise blood lactate (or surrogate) to support “greater oxygen debt” claims. Consider adding EPOC terminology for Post-EX and, if available, lactate or HR recovery data. And You conclude BC did not offset BFR’s effect on RPE. Given the RPE prescription confound, please soften this claim: at minimum, state that under a load titrated in the BFR+BC condition, average RPE during sprints was lower in OFF, and no independent effect of BC can be isolated without a factorial design.
Response 11: Our current protocol may have included varying resistance levels for each sprint, depending on participants’ RPE, resulting in individual differences in resistance profiles. Given the small sample size, we were unable to perform additional comparisons (rate traces); however, we have reported the mean rate by normalizing the total value by weight and time. The results have been updated accordingly. Furthermore, BC and BFR were also applied during the active rest periods; therefore, we considered the entire 21-minute protocol as a single integrated intervention. These limitations have been acknowledged in the Discussion section. We have also noted additional limitations of our study (e.g., lack of lactate and heart rate measurements) in the Discussion. The Conclusion has been revised as suggested, and references to EPOC have been incorporated into the Discussion, which has been expanded to reflect the reviewer’s recommendations. Please see the highlighted text in Results and Discussion.

Reviewer 3 Report
Comments and Suggestions for Authors
- The introduction presented two strategies with distinct physiological intentions: Blood Flow Restriction (BFR), which is more closely associated with the maximization of anaerobic responses under hypoxic conditions, and Body Cooling (BC), primarily aimed at exercise recovery. However, the introduction lacked a clear physiological explanation supporting their potential integration or competition. It remains unclear whether these interventions could operate synergistically or whether one might counteract or inhibit the effects of the other, which makes the overall justification of the proposal unconvincing. Furthermore, the introduction was somewhat biased toward emphasizing its application in individuals with persistent post-concussive symptoms (PPCS). This approach created a misleading expectation for the reader, as the study was ultimately conducted in healthy participants. This misalignment between the justification and the actual study population undermines the clarity of the introduction and should be carefully revised.
- In the Methods section, it should be explicitly stated that the compression applied to the arms (40 mmHg) and legs (65 mmHg) was maintained at the same levels under both conditions (cold-water and thermoneutral-water immersion). This clarification is essential to ensure that any differences in outcomes cannot be attributed to variations in cuff pressure. Additionally, it would be important to consider whether water temperature might influence cuff pressure stability. For instance, cold-induced vasoconstriction could alter limb circumference and, consequently, the actual pressure exerted by the cuffs. Addressing this aspect would enhance the methodological transparency and robustness of the study.
- It should also be explicitly reported in the Methods section what the water temperature was (in degrees Celsius) for each condition.
- It is unclear whether the six 30-second sprint intervals were performed at true maximal intensity. Was there a preliminary test to determine the expected power output for a single sprint? If the sprint intervals were indeed maximal, the term Sprint Interval Training (SIT) may be more appropriate than High-Intensity Training (HIT).
- Figure 3 is clear, but it could be improved for better visualization. I suggest including the effect size within each comparison to provide more insight into the magnitude of the differences.
- The main limitation in your results is the lack of mechanical data on the power output achieved during the sprints, which should be compared between the VASP ON and VASP OFF conditions. This information is crucial because, hypothetically, if it shows that power output in the VASP OFF condition was lower compared to VASP ON, the interpretation of the entire study could change. In that case, it would suggest that the ventilatory responses were influenced more by mechanical differences than by the VASP itself, which would fundamentally alter the conclusions. In summary, without the inclusion of mechanical data, I cannot recommend the approval of this manuscript.
Author Response
Comment 1: The introduction presented two strategies with distinct physiological intentions: Blood Flow Restriction (BFR), which is more closely associated with the maximization of anaerobic responses under hypoxic conditions, and Body Cooling (BC), primarily aimed at exercise recovery. However, the introduction lacked a clear physiological explanation supporting their potential integration or competition. It remains unclear whether these interventions could operate synergistically or whether one might counteract or inhibit the effects of the other, which makes the overall justification of the proposal unconvincing. Furthermore, the introduction was somewhat biased toward emphasizing its application in individuals with persistent post-concussive symptoms (PPCS). This approach created a misleading expectation for the reader, as the study was ultimately conducted in healthy participants. This misalignment between the justification and the actual study population undermines the clarity of the introduction and should be carefully revised.
Response 1: We appreciate the reviewer’s comment and have revised the Introduction to avoid emphasizing a single clinical population. In addition, we have expanded the Introduction to more clearly articulate the physiological rationale for the combined use of blood flow restriction and body cooling. Please see the highlighted text in the Introduction section.
Comment 2: In the Methods section, it should be explicitly stated that the compression applied to the arms (40 mmHg) and legs (65 mmHg) was maintained at the same levels under both conditions (cold-water and thermoneutral-water immersion). This clarification is essential to ensure that any differences in outcomes cannot be attributed to variations in cuff pressure. Additionally, it would be important to consider whether water temperature might influence cuff pressure stability. For instance, cold-induced vasoconstriction could alter limb circumference and, consequently, the actual pressure exerted by the cuffs. Addressing this aspect would enhance the methodological transparency and robustness of the study.
Response 2: In our study, cuff pressure was generated by filling the cuffs with chilled water. The Vasper system continuously monitors both water temperature and cuff pressure. We have revised the description of the method to improve clarity. Please refer to Lines 129-150.
Comment 3: It should also be explicitly reported in the Methods section what the water temperature was (in degrees Celsius) for each condition.
Response 3: We have provided the temperature information in degrees Celsius in the Methods section. Both conditions were conducted in the same room. The VASPER ON condition included local cooling (per-cooling, i.e., cooling during exercise) delivered through the Vasper system’s BC unit, which circulates chilled water through the cuffs, seat pad, and copper plates. The VASPER OFF condition did not include this local cooling during exercise. We have also included the environmental condition of the room as follows.
“The room where the exercise occurred was air-conditioned, with the thermostat set at 20°C. However, because the system was centrally controlled within a building, the actual temperature was approximately 25°C, about 2~5°C higher than the temperature recommended by the American College of Sports Medicine (ACSM) and the International Fitness Association (IFA), but still within the thermoneutral range. Participants did not express any discomfort or report feeling warm before exercise began. The average humidity was about 52%.”
Comment 4: It is unclear whether the six 30-second sprint intervals were performed at true maximal intensity. Was there a preliminary test to determine the expected power output for a single sprint? If the sprint intervals were indeed maximal, the term Sprint Interval Training (SIT) may be more appropriate than High-Intensity Training (HIT).
Response 4: Thank you for the comment. Moderate-intensity exercise was used in this study, and we have revised the Methods section to improve clarity. The protocol remained identical for both conditions to ensure that the total mechanical workload (21 minutes) was comparable. We have also added a post hoc data analysis, and the results indicate no significant difference in mechanical output between the two conditions among participants (p = 0.258, CI [−5963.71, 1678.19], d = −0.23). Accordingly, we have revised the Methods and Results sections to emphasize the importance of maintaining comparable mechanical output between the two conditions.
“The participant had to reach the target RPE scale level of 6 (higher end of moderate intensity) during the sprint intervals in VASPER ON condition. If the sprint interval was deemed too easy, the next sprint level’s resistance was increased by one level. The resistance levels of sprint intervals decided in VASPER ON were saved and used for VASPER OFF condition for each individual participant.” (Lines 177-181)
“To ensure the comparable mechanical work generated in two conditions, we also compared the sum of work (in Watts).” (Lines 218-219)
“The average mechanical outputs over 21 minutes for the VASPER ON and VASPER OFF conditions were 89909.56±12090.75 and 92052.32±8678.07, respectively. The correlation between the two conditions was statistically significant (r=0.65, p<0.001). There was no significant difference in mechanical output between the two conditions (p=0.258, CI [-5963.71 1678.19], d=-0.231). These results suggest that participants produced consistent mechanical outputs across both conditions.” (Lines 231-236)
Comment 5: Figure 3 is clear, but it could be improved for better visualization. I suggest including the effect size within each comparison to provide more insight into the magnitude of the differences.
Response 5: We have provided effect sizes, 95% CIs, and p-values. We have also included adjusted α values in Table 2.
Comment 6: The main limitation in your results is the lack of mechanical data on the power output achieved during the sprints, which should be compared between the VASP ON and VASP OFF conditions. This information is crucial because, hypothetically, if it shows that power output in the VASP OFF condition was lower compared to VASP ON, the interpretation of the entire study could change. In that case, it would suggest that the ventilatory responses were influenced more by mechanical differences than by the VASP itself, which would fundamentally alter the conclusions. In summary, without the inclusion of mechanical data, I cannot recommend the approval of this manuscript.
Response 6: We thank the reviewer for this important comment. As mentioned in the previous response (Response 4), in our study, we controlled the mechanical output across the two conditions by using the same resistance levels and performance feedback in both conditions. Participants were required to generate sufficient power to maintain their performance within a target window displayed on the tablet screen (Fig. 1). Although a counterbalanced order would have been ideal, preliminary testing indicated that the resistance levels established during the VASPER OFF condition could not be maintained when the VASPER ON condition was performed. We have also acknowledged limitations in the Discussion section (Lines391-402).
“The resistance levels of sprint intervals decided in VASPER ON were saved and used for VASPER OFF condition for each individual participant. The VASPER ON was conducted first to determine the appropriate resistance for the moderate intensity workout, based on the assumption that it would be the more demanding of the two conditions. This approach allowed for adjustments to the sprint intervals to be made and saved, ensuring that participants were able to complete the identical protocol in both conditions. Using feedback displayed on the tablet screen (Fig. 1), participants had to generate preset Watts determined by the resistance level. Because the resistance levels remained the same in two conditions, the mechanical output remained comparable. No resistance level changes were made to the warm-up and rest intervals of the protocol.” (Lines 182-189 & Fig 1 legend)

Reviewer 4 Report
Comments and Suggestions for Authors
Short-Term Effects of an Exercise Protocol Incorporating Blood Flow Restriction and Body Cooling in Healthy Young Adults
This study investigates the acute physiological and perceptual responses to interval training with concurrent blood flow restriction and body cooling (VASPER ON) compared to a control condition (VASPER OFF). The topic is relevant and innovative, given the recent surge in interest around combined recovery and hypoxic-like interventions. The manuscript is generally well-written and methodologically clear. However, there are conceptual gaps, analytical limitations, and interpretative inconsistencies that should be addressed to strengthen the scientific rigor and theoretical contribution of the study.
MAJOR COMMENTS
Rationale
The introduction appropriately contextualizes BFR and BC independently but does not clearly establish a physiological or mechanistic rationale for their combined use. For example, it is unclear how cooling might modulate hemodynamic or metabolic responses induced by BFR—whether expected to amplify or attenuate them. Please add a short paragraph explaining the potential interactive mechanisms (e.g., cooling-induced vasoconstriction vs. BFR-induced ischemia) and how these may influence oxygen delivery, CO₂ removal, or autonomic regulation during and after exercise.
The authors mention public health and exercise intolerance (PPCS) but the link between PPCS and the current healthy cohort is weak. If this protocol aims to inform future applications for PPCS, that translational logic should be briefly reinforced.
Experimental Design and Physiological Measurements
The within-subject crossover design is appropriate, but randomization and counterbalancing are not mentioned. Performing VASPER ON first for all participants introduces a strong order effect and possible learning/adaptation bias. Explicitly acknowledge this limitation and discuss how it might have influenced metabolic or perceptual outcomes.
It is unclear whether participants were blinded to the hypothesis or to the purpose of cooling and compression. Expectation bias could influence RPE and ventilatory responses.
Environmental conditions (ambient temperature, humidity) during testing should be reported, as they critically affect thermoregulatory and ventilatory variables, particularly when cooling is part of the intervention.
Cuff pressure values (40 and 65 mmHg) are relatively low for eliciting significant arterial occlusion; provide justification for these magnitudes, ideally referencing studies that have shown metabolic perturbation under similar pressures.
The analysis reports only aggregate totals (TVO₂, TVCO₂, TVE, etc.) for 21 min of exercise and 10 min post-exercise. This approach masks temporal dynamics across intervals and recovery. Please include (or at least mention) whether time-course data were examined (e.g., minute-by-minute or interval-by-interval trends). This could reveal meaningful physiological patterns obscured by total sums.
The absence of heart rate, skin temperature, or blood lactate data limits interpretation of the metabolic stress imposed by BFR+BC. While these measures may not have been collected, their omission should be noted as a limitation, as they are direct markers of physiological demand.
The discussion states that post-exercise oxygen consumption increased due to “greater oxygen debt,” but this claim would benefit from a quantitative comparison (e.g., relative percent increase) and a reference to EPOC (excess post-exercise oxygen consumption) literature.
Statistical Approach
Using paired t-tests/Wilcoxon tests for multiple variables (six during EX, six during Post-EX) without correction for multiple comparisons (e.g., Bonferroni, FDR) inflates the type I error rate. Please apply or at least discuss a correction for multiple testing.
The paper reports p-values only, with no effect sizes or confidence intervals. Given the small sample (n = 26), effect sizes are essential to interpret the magnitude of physiological differences. It is essential to report Cohen’s d (or r for nonparametric data) and 95% CI for key outcomes.
The manuscript acknowledges 26 participants but provides no priori power calculation or effect-size rationale. Add at least a retrospective estimation (using observed SDs) to clarify whether the study was powered to detect small-to-moderate physiological differences.
The use of total oxygen consumption in mL (TVO₂) is unconventional; it would be more interpretable to normalize to body mass and express as mL·kg⁻¹·min⁻¹, facilitating comparison with prior literature.
Discussion/Data Interpretation
The authors attribute elevated ventilation to CO₂ accumulation in venous blood during BFR; this is plausible but not directly evidenced in the present study. Strengthen this section by referencing literature on chemoreceptor sensitivity or respiratory control during hypoxia/hypercapnia under BFR (e.g., Hunt et al., J Appl Physiol, 2016).
Also, the discussion could benefit from explicitly distinguishing acute ventilatory drive mechanisms (neural, chemical) from metabolic post-exercise responses (EPOC, lactate oxidation, thermoregulation). Currently, these are intertwined, reducing conceptual clarity.
The claim that BC did not offset the effects of BFR should be substantiated with more detailed reasoning—e.g., that cooling’s vasoconstrictive effect may have compounded ischemic stress rather than attenuating it.
MINOR COMMENTS
Use consistent terms: “exercise” vs. “training session,” “oxygen uptake” vs. “oxygen consumption,” etc.
Figures are clear but lack units on y-axes in several panels.
The use of “†” for significance is fine, but legends should specify the exact p-value threshold (e.g., “† p < 0.05”).
Abstract:
The last sentence could better reflect the main conclusion: “These findings suggest that concurrent blood flow restriction and body cooling increase post-exercise metabolic and ventilatory demands without attenuating each other’s effects.”
Comments on the Quality of English LanguageThe English could be refined in certain sections to enhance conciseness and clarity.
Author Response
Comment 1: The introduction appropriately contextualizes BFR and BC independently but does not clearly establish a physiological or mechanistic rationale for their combined use. For example, it is unclear how cooling might modulate hemodynamic or metabolic responses induced by BFR—whether expected to amplify or attenuate them. Please add a short paragraph explaining the potential interactive mechanisms (e.g., cooling-induced vasoconstriction vs. BFR-induced ischemia) and how these may influence oxygen delivery, CO₂ removal, or autonomic regulation during and after exercise.
Response 1: We have expanded the Introduction to more clearly articulate the physiological rationale for the combined use of blood flow restriction and body cooling.
Comment 2: The authors mention public health and exercise intolerance (PPCS) but the link between PPCS and the current healthy cohort is weak. If this protocol aims to inform future applications for PPCS, that translational logic should be briefly reinforced.
Response 2: We appreciate the reviewer’s comment and have revised the Introduction to avoid emphasizing a single clinical population.
Comment 3: The within-subject crossover design is appropriate, but randomization and counterbalancing are not mentioned. Performing VASPER ON first for all participants introduces a strong order effect and possible learning/adaptation bias. Explicitly acknowledge this limitation and discuss how it might have influenced metabolic or perceptual outcomes. It is unclear whether participants were blinded to the hypothesis or to the purpose of cooling and compression. Expectation bias could influence RPE and ventilatory responses.
Response 3: Although a counterbalanced order would have been ideal, preliminary testing showed that resistance levels established during the VASPER OFF condition could not be achieved when VASPER ON was performed. To ensure comparable mechanical work between conditions, we conducted VASPER ON first to determine the appropriate resistance and then replicated the same workload (same levels of resistance) for VASPER OFF. This approach allowed us to control exercise intensity across both conditions so that any observed physiological differences could be attributed to the intervention rather than differences in workload (mechanical output/power). Participants were not blinded to the exercise protocols; however, they were blinded to the study hypothesis and any potential effects the protocols might produce. The revised manuscript includes additional details on mechanical output and clarifies this decision in the Methods section. We have also acknowledged the limitations in the Discussion.
“The VASPER ON was conducted first to determine the appropriate resistance for the moderate intensity workout, based on the assumption that it would be the more demanding of the two conditions. This approach allowed for adjustments to the sprint intervals to be made and saved, ensuring that participants completed the identical protocol in both conditions.” (Lines 182-186)
“All participants reported exercising regularly at least twice a week and had no experience with BFR or BC. When it was not possible to blind the participants to the exercise protocol, they were not informed of any potential effects that could arise from either of the two protocols they were to perform.” (Lines 121-123)
Comment 4: Environmental conditions (ambient temperature, humidity) during testing should be reported, as they critically affect thermoregulatory and ventilatory variables, particularly when cooling is part of the intervention.
Response 4: We did not precisely control the environmental conditions; however, both exercise protocols were conducted in the same air-conditioned room and at approximately the same time of day. A corresponding statement has been added to the Methods section as follows.
“The room where the exercise occurred was air-conditioned, with the thermostat set at 20°C. However, because the system was centrally controlled within a building, the actual temperature was approximately 25°C, about 2~5°C higher than the temperature recommended by the American College of Sports Medicine (ACSM) [27] and the International Fitness Association (IFA), but still within the thermoneutral range. Participants did not express any discomfort or report feeling warm before exercise began. The average humidity was about 52%.” (Lines 199-206)
Comment 5: Cuff pressure values (40 and 65 mmHg) are relatively low for eliciting significant arterial occlusion; provide justification for these magnitudes, ideally referencing studies that have shown metabolic perturbation under similar pressures.
Response 5: The pressure values were selected based on previous literature and participant comfort. Laurentino et al. reported that moderate blood flow restriction BFR can elicit strength gains comparable to those achieved with high-intensity training without BFR, while producing less discomfort. Because BFR was applied to all four limbs in the present study, a moderate restriction level was chosen to examine potential acute physiological effects while maintaining tolerability. In their work, the pressure used for the lower extremities was approximately 50% of that typically applied in traditional BFR protocols. Laurentino et al. also noted that wider cuffs require lower pressures to achieve comparable levels of vascular occlusion; for instance, approximately 140 mmHg with an 18-cm cuff. We used cuffs of slightly greater width; therefore, pressures of 65 mmHg for the legs and 40 mmHg for the arms were selected to achieve moderate BFR and ensure participant comfort. These pressure levels were consistent with those used in our previous study involving individuals with persistent post-concussive symptoms; however, that study focused on clinical symptom improvement rather than acute physiological responses.
Cited reference: Laurentino, G. C., Ugrinowitsch, C., Roschel, H., Aoki, M. S., Soares, A. G., Neves, M., Jr., Aihara, A. Y., Fernandes Ada, R., & Tricoli, V. (2012). Strength training with blood flow restriction diminishes myostatin gene expression. Med Sci Sports Exerc, 44(3), 406-412. https://doi.org/10.1249/MSS.0b013e318233b4bc
“The pressure values used in this study were based on previous literature [27] and participant comfort. Laurentino et al. reported that moderate BFR (~50% of the typical pressure used) can elicit strength gains comparable to those achieved with high-intensity training without BFR, while producing less discomfort. They also noted that wider cuffs require lower pressures to achieve comparable levels of vascular occlusion. For instance, approximately 140 mmHg is required when using an 18-cm cuff to induce vascular occlusion. We used a slightly greater cuff width for the legs; therefore, we set the pressure at 65 mmHg for the legs. Although the cuff width for the arms was smaller than that used by Laurentino et al., the arm circumference is considerably smaller than that of the thighs; therefore, we selected 40 mmHg for the arms. These selected values were consistent with those used in our previous study [25].”
Comment 6: The analysis reports only aggregate totals (TVO₂, TVCO₂, TVE, etc.) for 21 min of exercise and 10 min post-exercise. This approach masks temporal dynamics across intervals and recovery. Please include (or at least mention) whether time-course data were examined (e.g., minute-by-minute or interval-by-interval trends). This could reveal meaningful physiological patterns obscured by total sums. The absence of heart rate, skin temperature, or blood lactate data limits interpretation of the metabolic stress imposed by BFR+BC. While these measures may not have been collected, their omission should be noted as a limitation, as they are direct markers of physiological demand.
Response 6: Our current protocol may have included varying resistance levels for each sprint, depending on participants’ RPE, resulting in individual differences in resistance profiles. Given the small sample size, we were unable to perform additional comparisons. Furthermore, BC and BFR were also applied during the active rest periods; therefore, we considered the entire 21-minute protocol as a single integrated intervention. These limitations have been acknowledged in the Discussion section.
Comment 7: The discussion states that post-exercise oxygen consumption increased due to “greater oxygen debt,” but this claim would benefit from a quantitative comparison (e.g., relative percent increase) and a reference to EPOC (excess post-exercise oxygen consumption) literature.
Response 7: We did not record resting-state oxygen consumption; therefore, we are unable to provide a quantitative comparison. However, based on our observations, there was no significant difference in oxygen consumption between the two conditions during exercise, while higher values were observed during the post-exercise period. Given that RPE was higher in the VASPER ON condition, we would have expected greater oxygen consumption during exercise. Since this was not observed, we speculate that insufficient oxygen availability may have resulted in some oxygen “debt,” leading to elevated post-exercise oxygen consumption. This has been addressed in the revised Discussion section.
Comment 8: Using paired t-tests/Wilcoxon tests for multiple variables (six during EX, six during Post-EX) without correction for multiple comparisons (e.g., Bonferroni, FDR) inflates the type I error rate. Please apply or at least discuss a correction for multiple testing. The paper reports p-values only, with no effect sizes or confidence intervals. Given the small sample (n = 26), effect sizes are essential to interpret the magnitude of physiological differences. It is essential to report Cohen’s d (or r for nonparametric data) and 95% CI for key outcomes.
Response 8: Thank you for the suggestion. We have applied Benjamini-Hochberg approach to correct the multiple comparison and reported the findings in Table 2. We have also reported effect size and 95% CI in the results. We have also included effect sizes and 95% Cis.
Comment 9: The manuscript acknowledges 26 participants but provides no priori power calculation or effect-size rationale. Add at least a retrospective estimation (using observed SDs) to clarify whether the study was powered to detect small-to-moderate physiological differences.
Response 9: We have added information to the method (section 2.1) as follows.
“Although an a priori power analysis was not performed, a post hoc power analysis using G*Power 3.1 indicated that the sample size of 26 participants yielded 91% power to detect an effect size of 0.69, based on the observed difference in RPE responses between the two conditions.”
Comment 10: The use of total oxygen consumption in mL (TVO₂) is unconventional; it would be more interpretable to normalize to body mass and express as mL·kg⁻¹·min⁻¹, facilitating comparison with prior literature.
Response 10: Thank you for the suggestion. We have updated the manuscript to report TVO₂ and TVCO₂ in mL·kg⁻¹·min⁻¹, normalized to body mass, to align with standard reporting conventions.
Comment 11: The authors attribute elevated ventilation to CO₂ accumulation in venous blood during BFR; this is plausible but not directly evidenced in the present study. Strengthen this section by referencing literature on chemoreceptor sensitivity or respiratory control during hypoxia/hypercapnia under BFR (e.g., Hunt et al., J Appl Physiol, 2016). Also, the discussion could benefit from explicitly distinguishing acute ventilatory drive mechanisms (neural, chemical) from metabolic post-exercise responses (EPOC, lactate oxidation, thermoregulation). Currently, these are intertwined, reducing conceptual clarity. The claim that BC did not offset the effects of BFR should be substantiated with more detailed reasoning—e.g., that cooling’s vasoconstrictive effect may have compounded ischemic stress rather than attenuating it.
Response 11: We have revised the introduction to better separate the effects observed during exercise and post exercise. We expand the discussion on the potential mechanisms. Please see the highlighted text in the Discussion section.
Comment 12: Use consistent terms: “exercise” vs. “training session,” “oxygen uptake” vs. “oxygen consumption,” etc.
According to the reviewer’s suggestions, we have revised the terminology to use ‘exercise’ when describing our testing conditions (VASPER OFF and VASPER ON) and reserved ‘training’ (used with ‘interval’) exclusively for interval training.
Response 12: We have also used ‘oxygen consumption’ consistently throughout the manuscript.
Comment 13: Figures are clear but lack units on y-axes in several panels.
Response 13: The y-axes without units represent either dimensionless variables (e.g., VE/VCO₂, VE/VO₂) or ordinal data (e.g., RPE and resistance level in Figure 2). To avoid confusion, we have clarified that the y-axis for resistance represents the exercise machine’s resistance level, which is on an ordinal scale where higher numbers indicate greater resistance in Fig 2 legend.
“Figure 2. … The resistance levels provided by the exercise machine are on an ordinal scale where higher numbers indicate greater resistance.”
Comment 14: The use of “†” for significance is fine, but legends should specify the exact p-value threshold (e.g., “† p < 0.05”).
Response 14: We have revised the legend as follows.
“The dagger (†) indicates statistically significant differences between two conditions after comparing the p values to the adjusted α level using Benjamini-Hochberg test.”
Comment 15: Abstract: The last sentence could better reflect the main conclusion: “These findings suggest that concurrent blood flow restriction and body cooling increase post-exercise metabolic and ventilatory demands without attenuating each other’s effects.”
Response 15: We have revised and toned down the last sentence as suggested.

Round 2
Reviewer 2 Report
Comments and Suggestions for Authors
Dear authors,
well done
Author Response
We sincerely appreciate the reviewer's continued support and the valuable comments provided earlier, which have significantly improved the quality of our manuscript.
Reviewer 3 Report
Comments and Suggestions for Authors
After rereading the manuscript, I believe the authors have considerably improved several aspects. However, I have one additional comment that I noticed during this new reading. It seems that the initial argument (at the beginning of the introduction), which aims to identify novel exercise strategies that make physical activity more tolerable for individuals with medical conditions, appears contradictory to the final statement of the introduction, which suggests that exercising under combined BC and BFR potentially leads to greater perceived exertion. In this context, such a combination could even be contraindicated for individuals with limited exercise tolerance. Nevertheless, I understand the authors’ point, which could be clarified or strengthened by better addressing the role of exercise intensity in this context.
Author Response
Comment 1: It seems that the initial argument (at the beginning of the introduction), which aims to identify novel exercise strategies that make physical activity more tolerable for individuals with medical conditions, appears contradictory to the final statement of the introduction, which suggests that exercising under combined BC and BFR potentially leads to greater perceived exertion. In this context, such a combination could even be contraindicated for individuals with limited exercise tolerance. Nevertheless, I understand the authors’ point, which could be clarified or strengthened by better addressing the role of exercise intensity in this context.
Response 1: We sincerely thank the reviewer for this insightful comment. We agree that the initial rationale in the introduction may have appeared inconsistent with the concluding speculative statement, and we appreciate the opportunity to clarify this point. Our main objective was to investigate the acute effects of combining BC and BFR during interval training, given that both techniques independently offer distinct physiological benefits. Although we speculated that exercising under combined BC and BFR conditions may increase perceived exertion, this response may emulate the stimulus of more intensive exercise (e.g., longer duration or higher load) while still being tolerable and achievable for participants. Thus, the combination could potentially confer similar benefits to high-intensity exercise, even in populations with limited exercise tolerance. We have revised the last paragraph of the introduction as follows:
“Despite our speculation, this response may replicate the stimulus of more demanding exercise while remaining within tolerable limits, as the combined application of BFR and BC introduces complexities that are not yet understood.” (Lines 99-101)
